behaviour

temperament, behavioural syndrome, ontogeny, development

**Author for correspondence:**
Marina A. G. von Keyserlingk
e-mail: nina@mail.ubc.ca

†Present address: AgResearch Ltd., Ruakura Research Centre, 10 Bisley Road, Hamilton 3214, New Zealand.
‡Present address: Dairy Science Program, Department of Animal and Food Sciences, University of Kentucky, 325 Cooper Drive, Lexington 40546, KY, USA.

# Long-term consistency of personality traits of cattle

Heather W. Neave†, Joao H. C. Costa‡, Daniel M. Weary and Marina A. G. von Keyserlingk

Animal Welfare Program, Faculty of Land and Food Systems, University of British Columbia, Vancouver, British Columbia, Canada V6T 1Z4

HWN, 0000-0002-1818-8131; JHCC, 0000-0001-9311-4741; DMW, 0000-0002-0917-3982; MAGvK, 0000-0002-1427-3152

Personality is often defined as the behaviour of individual animals that is consistent across contexts and over time. Personality traits may become unstable during stages of ontogeny from infancy to adulthood, especially during major periods of development such as around the time of sexual maturation. The personality of domesticated farm animals has links with productivity, health and welfare, but to our knowledge, no studies have investigated the development and stability of personality traits across developmental life stages in a mammalian farm animal species. Here, we describe the consistency of personality traits across ontogeny in dairy cattle from neonate to first lactation as an adult. The personality traits 'bold' and 'exploratory', as measured by behavioural responses to novelty, were highly consistent during the earlier (before and after weaning from milk) and later (after puberty to first lactation) rearing periods, but were not consistent across these rearing periods when puberty occurred. These findings indicate that personality changes in cattle around sexual maturation are probably owing to major physiological changes that are accelerated under typical management conditions at this time. This work contributes to the understanding of the ontogeny of behaviour in farm animals, especially how and why individuals differ in their behaviour.

## 1. Introduction

Personality is often defined as individual differences in behaviour that are consistent across time and situations [1]. However, personality traits may not be consistent across some stages of ontogeny from juvenile to adult, especially during major periods of development such as morphogenesis, sexual maturation or times of significant neurological and physiological reorganization [2,3]. For instance, changes in personality were observed at or after sexual maturation in humans [4] and a number of other animal species across taxa (e.g. stickleback fish [5], squid [6],

guinea pigs [7], marmots [8], junglefowl [9]). However, species such as the rat [10], squirrel [11], lake frog [12] and damselfly [13] showed stable personality traits (such as boldness, exploration or activity) from juvenile to adult stages. Therefore, the life stage at which adult personality is established appears to differ among species. The reason for differences in stability of personality across ontogeny may be related to changes in internal factors such as growth rate, hormone profile, metabolism and morphology, or to changes in physical or social environments [14].

Here, we aim to describe the development and stability of personality traits during ontogeny of Holstein dairy cattle from calf to adulthood. There is growing interest in the study of personality of domesticated ruminants owing to its links with individual productivity, health and welfare [15,16], yet there are only a few studies investigating the development and stability of personality traits across life stages in ruminant species. The most common personality traits investigated in farm animals include 'boldness' (i.e. the propensity to take risks when exposed to novelty [17]), 'exploration' (i.e. behaviour that gains information about the environment [18]) and 'sociability' (i.e. propensity to seek contact with or remain close to conspecifics [1]); these traits are typically inferred by observing behavioural responses to novelty or isolation in standardized tests [19]. Studies of dairy cattle have shown short-term consistency in individual responses towards a novel human or object as calves (measured three weeks apart before two months of age; [20]) and as lactating adults (measured six months apart; [21]). Others have reported consistency in these behavioural responses over longer periods of time, such as from three weeks to six months of age [22,23], from 3 to 7 months of age [24] and from 7 to 24 months of age [25]. However, no work to our knowledge has investigated the consistency of individual behavioural responses to novelty, and stability of personality traits of dairy cattle from an early age to adulthood. Some work in wild ruminants showed reproductive status and age did not affect the stability of boldness and docility traits (bighorn ewes: [26]). Dairy cattle provide an opportunity to investigate the development and consistency of behaviour because these animals undergo several important developmental and physiological changes that may be accelerated under commercial management conditions including immediate separation from the dam at birth, weaning from milk to solid feed at around two months of age [27], onset of puberty before 12 months of age [28], pregnancy and onset of lactation after calving typically by 24 months of age. The study of behavioural patterns and personality traits across these transitions may offer insight into the factors affecting the stability of personality across ontogeny, although we do not aim to specifically test the influence of each of these factors. Our study also included a sub-sample of male calves that were only studied during the early rearing period; this provided an opportunity to investigate sex differences in the expression of personality traits in dairy calves.

# 2. Material and methods

## 2.1. Animal management

This experiment used two cohorts of Holstein calves (cohort 1: $n = 33$ female calves; cohort 2: $n = 32$ female calves) that were studied longitudinally from pre-weaning to first lactation. Cohort 2 also included $n = 24$ male calves that were studied from pre-weaning to post-weaning. All animals were reared on the same farm and management of animals was the same for all periods except for the pre-weaning period. Calves in cohort 1 (born November 2014 to February 2015) received $12 \, l \, d^{-1}$ of milk for six weeks then weaned over two weeks to complete weaning at eight weeks, while calves in cohort 2 received different milk allowances (6, 8, 10 or $12 \, l \, d^{-1}$ of milk) and weaned on a step-down weaning programme reported in Neave *et al.* [29] (milk allowance did not affect measures of personality, see Neave *et al.* [29]). After the post-weaning period, all animals were managed according to standard farm management procedures. Some female calves and all male calves were sold after weaning (following common practice on this farm) and thus were unavailable for final testing, resulting in 22 of the original 33 female calves in cohort 1 and 26 of the original 32 female calves in cohort 2 completing the study.

## 2.2. Personality tests

Animals were subjected individually to three tests designed to measure behavioural reactivity to novelty at four different developmental periods: pre-weaning (age $27 \pm 3 \, d$ for cohort 2: $n = 56$), post-weaning (age $100 \pm 12 \, d$ for cohort 1: $n = 33$; age $76 \pm 3 \, d$ for cohort 2: $n = 56$), after puberty (hereafter called 'puberty'; age $11.7 \pm 0.4$ months for cohort 1: $n = 30$) and during first lactation after confirmation of

second pregnancy (hereafter called 'lactation'; age 29.6 ± 2.9 months for cohort 1: $n = 22$; age 29.6 ± 1.9 months for cohort 2: $n = 26$). Cohort 1 was not tested in the pre-weaning period (owing to a convenience sample that was enrolled at post-weaning only), and cohort 2 was not tested in the puberty period (owing to time and barn space constraints).

Personality tests conducted at each period were novel environment (empty arena, 30 min), novel human (unfamiliar person at arena centre, 10 min) and novel object tests (unfamiliar object at arena centre, 15 min), conducted individually over three consecutive days in a testing arena that was visually isolated from the herd. Behaviours during each test were recorded continuously. A single observer scored all behaviours in all tests following Neave et al. [29] using a detailed ethogram (see the electronic supplementary material, table S1).

## 2.3. Statistical analysis

Data were analysed with SAS (v. 9.4; SAS Inst. Inc., Cary, NC, USA), using the individual animal as the experimental unit. To determine the relationship between individual behavioural measurements across periods, Spearman rank correlations were performed owing to the non-parametric nature of most variables.

A principal component analysis (PCA) was used to condense behavioural measurements from the novelty tests into principal components (factors) for each period (i.e. a separate PCA was performed for each of the pre-weaning, post-weaning, puberty and lactation periods, following Neave et al. [29]). Final input variables from the novel human and object tests included: latency to touch the human and object, time spent touching the human and object, time spent attentive (looking at the human and object) and inattentive, and number of object play events. Final input variables from the novel environment test included time spent exploring, active and inactive. For each PCA, the first two factors were retained and subjected to varimax rotation. Factor loadings for each animal from each PCA were extracted. Pearson correlations were performed between the individual factor loadings at each period to determine if the correlational structure of personality traits changes over time (i.e. structural consistency). Tucker's coefficient of congruence was also performed as a measure of overall factor loading similarity (i.e. similar correlated sets of behaviours across periods, and thus structural consistency) across periods (following Lorenzo-Seva and Ten Berge [30]).

To determine if male and female calves differed in their expression of personality traits (i.e. individual factor loadings), we conducted two general linear models where factor loading at each period was the response variable (pre-weaning and post-weaning periods only, because males were only available at these periods) and sex and birth weight were the explanatory variables. To determine if male and female calves differed in the consistency of personality traits from pre-weaning to post-weaning, a general linear model was conducted where post-weaning factor loading was the response variable, and pre-weaning factor loading, sex, birth weight and the interaction of pre-weaning factor loading and sex were explanatory variables.

See the electronic supplementary material for more information on materials and methods, statistical analysis and datasets.

# 3. Results

## 3.1. Rank-order consistency of behavioural measures

Descriptive behavioural responses in each of the novelty tests (mean, s.d. and range) for each period (pre-weaning, post-weaning, puberty and lactation) are presented in the electronic supplementary material, table S2.

Behaviours expressed in the novelty tests correlated with the same behaviours expressed in the same tests at later periods, especially between pre- and post-weaning, and between puberty and lactation (table 1). All behaviours expressed in the novel environment test were correlated (or tended to be correlated) between puberty and lactation periods, but were less consistent between other periods. All behaviours expressed in the novel human test were correlated between pre-weaning and post-weaning periods, and most behaviours were also correlated between puberty and lactation periods. Some behaviours in the novel human test were correlated between post-weaning and puberty periods, but there were no correlations between pre-weaning and lactation, or between post-weaning and lactation periods. Most behaviours expressed in the novel object test were correlated between pre-weaning and post-weaning, and between puberty and lactation, but there were no (or limited) correlations between pre-weaning and lactation, post-weaning and puberty, and post-weaning and lactation.

**Table 1.** Spearman rank correlations (*r*) of individual behavioural measures of Holstein dairy cattle in three novelty tests (novel environment, novel human and novel object) across four developmental periods (pre-weaning, post-weaning, puberty and lactation). (Significant correlations are italicized ($p < 0.05$) and tendencies are in bold ($0.05 < p < 0.10$).)

| test/behaviour | pre-weaning to post-weaning ($n = 56$) | | pre-weaning to lactation ($n = 26$) | | post-weaning to puberty ($n = 29$) | | post-weaning to lactation ($n = 48$) | | puberty to lactation ($n = 21$) | |
|---|---|---|---|---|---|---|---|---|---|---|
| | *r* | *p*-value | *r* | *p*-value | *r* | *p*-value | *r* | *p*-value | *r* | *p*-value |
| novel environment test | | | | | | | | | | |
| exploration (% of test time) | −0.05 | 0.74 | −0.13 | 0.53 | 0.09 | 0.66 | **0.27** | **0.07** | 0.52 | 0.02 |
| inactivity (% of test time) | 0.08 | 0.53 | 0.31 | 0.12 | 0.46 | 0.01 | 0.19 | 0.19 | 0.61 | 0.004 |
| locomotory play (no.) | 0.10 | 0.45 | −0.03 | 0.88 | 0.14 | 0.50 | 0.18 | 0.23 | 0.25 | 0.26 |
| active (no. quadrants crossed) | −0.04 | 0.78 | 0.49 | 0.01 | 0.22 | 0.26 | 0.23 | 0.13 | 0.58 | 0.006 |
| novel human test | | | | | | | | | | |
| latency to touch (s) | 0.50 | <0.001 | 0.02 | 0.93 | 0.42 | 0.02 | 0.06 | 0.70 | **0.42** | **0.06** |
| attentive (% of test time) | 0.53 | <0.001 | 0.31 | 0.14 | 0.29 | 0.12 | 0.29 | 0.05 | **0.38** | **0.09** |
| inattentive (% of test time) | 0.29 | 0.03 | 0.08 | 0.69 | 0.21 | 0.26 | 0.07 | 0.62 | 0.48 | 0.03 |
| touching human (% of test time) | 0.53 | <0.001 | −0.09 | 0.66 | 0.68 | <0.001 | 0.06 | 0.72 | 0.36 | 0.11 |
| locomotor play (no.) | 0.56 | <0.001 | 0.10 | 0.65 | 0.19 | 0.32 | −0.11 | 0.47 | 0.26 | 0.26 |
| novel object test | | | | | | | | | | |
| latency to touch (s) | 0.40 | 0.002 | 0.42 | 0.03 | −0.10 | 0.61 | 0.08 | 0.57 | **0.39** | **0.08** |
| attentive (% of test time) | 0.34 | 0.009 | 0.09 | 0.66 | −0.008 | 0.97 | 0.16 | 0.29 | 0.54 | 0.01 |
| inattentive (% of test time) | 0.09 | 0.53 | 0.14 | 0.50 | −0.11 | 0.56 | **0.26** | **0.07** | 0.63 | 0.002 |
| touching object (% of test time) | 0.18 | 0.20 | 0.03 | 0.89 | 0.13 | 0.50 | 0.05 | 0.73 | 0.49 | 0.02 |
| locomotor play (no.) | 0.30 | 0.03 | −0.14 | 0.36 | **0.31** | **0.10** | 0.31 | 0.03 | 0.30 | 0.19 |

## 3.2. Structural consistency of personality traits

The first two principal components (factors) were retained from each PCA. The total variation in behavioural measures explained by the factor, eigenvalues and loadings for each factor at each period are presented in the electronic supplementary material, table S2. For ease of interpretation and to be consistent with literature terminology, the loadings on some factors were reversed (factor 1 at post-weaning and puberty periods; factor 2 at puberty and lactation periods). In general, the first and second factors for each period described similar sets of correlated behaviours. We described the first factor as reflecting 'boldness', containing high negative loadings for latency to touch and attention towards the human or object, and high positive loadings for time spent touching and playing with the human or object. We described the second factor as reflecting exploration and activity (exploratory-active), containing high positive loadings for either exploration or activity, with some periods also having a high negative loading for inactivity.

Individual animals had consistent scores on factor 1 (bold) between pre-weaning and post-weaning ($r = 0.67$; $p < 0.001$; $n = 53$), between post-weaning and puberty ($r = 0.66$; $p < 0.001$; $n = 27$), and between puberty and lactation periods ($r = 0.48$; $p = 0.03$; $n = 21$), indicating that individual positioning along the 'boldness' axis was stable during these developmental periods (figure 1$a$,$c$,$e$). Individual scores on factor 1 were not consistent between pre-weaning and lactation ($r = -0.24$; $p = 0.23$; $n = 25$), or between post-weaning and lactation periods ($r = -0.05$; $p = 0.76$; $n = 44$) (figure 2$a$,$c$).

Animals also had consistent scores on factor 2 (exploratory-active) between post-weaning and puberty ($r = 0.40$; $p = 0.04$; $n = 27$), between puberty and lactation ($r = 0.49$; $p = 0.02$; $n = 21$), and tended to be consistent between pre-weaning and post-weaning periods ($r = 0.24$; $p = 0.08$; $n = 53$) (figure 1$b$,$d$,$f$), indicating that level of exploration or activity was stable during these developmental periods. Individual scores on factor 2 were not consistent between pre-weaning and lactation ($r = -0.19$; $p = 0.37$; $n = 25$), or between post-weaning and lactation periods ($r = -0.17$; $p = 0.28$; $n = 44$) (figure 2$b$,$d$).

Tucker's congruence coefficient indicated equal similarity of overall loadings (i.e. correlated sets of behaviours) on the first factor (bold) between pre-weaning and post-weaning (0.98) and between post-weaning and puberty (0.94), and high similarity between puberty and lactation periods (0.90). Moderately high similarity of loadings on the second factor (exploratory-active) was also found between puberty and lactation periods (0.80). This suggests that similar sets of correlated behaviours explained the majority of variation in behaviours during the novelty tests at these developmental periods.

## 3.3. Effect of sex

Males and females did not differ in their factor loadings on factor 1 (bold) (pre-weaning: $F_{1,53} = 0.01$, $p = 0.93$; post-weaning: $F_{1,50} = 0.17$, $p = 0.68$), or on factor 2 (exploratory-active) (pre-weaning: $F_{1,53} = 0.43$, $p = 0.52$; post-weaning: $F_{1,50} = 0.13$, $p = 0.72$). Consistency of personality traits between pre-weaning and post-weaning were similar for males and females (factor 1: $F_{1,48} = 0.51$, $p = 0.48$; factor 2: $F_{1,48} = 0.42$, $p = 0.52$).

## 4. Discussion

We investigated the long-term consistency of individual behavioural responses to novelty, and the stability of derived personality traits, over four periods of development; to our knowledge this is the first to study the ontogeny of personality traits from early age to adulthood in a commercial farm animal species. We found that cattle were consistent in their behavioural responses to novelty during the early rearing periods (from pre- to post-weaning), and during the later periods of development (from puberty to lactation). The personality trait 'boldness' was also highly consistent during these periods. However, individual consistency of behaviours and personality traits were generally poor across the major developmental period of puberty (from pre-weaning to lactation, and from post-weaning to lactation).

The underlying structure of the behavioural responses to each of the novelty tests comprised two main PCA components, which we interpreted as reflecting the personality traits 'boldness' and 'exploration–activity'. These traits have also been identified in dairy calves [20,22,24] and cows [21,31] with reasonable temporal stability using similar tests (although see [32] reporting low stability of behaviours in novelty tests over a short period of 1–3 weeks in calves). In our study, the 'boldness' trait had similarly high loadings for behaviours related to approach and contact with the novel human and object across the early (pre- to post-weaning) and later (puberty to lactation) periods of development; this suggests high

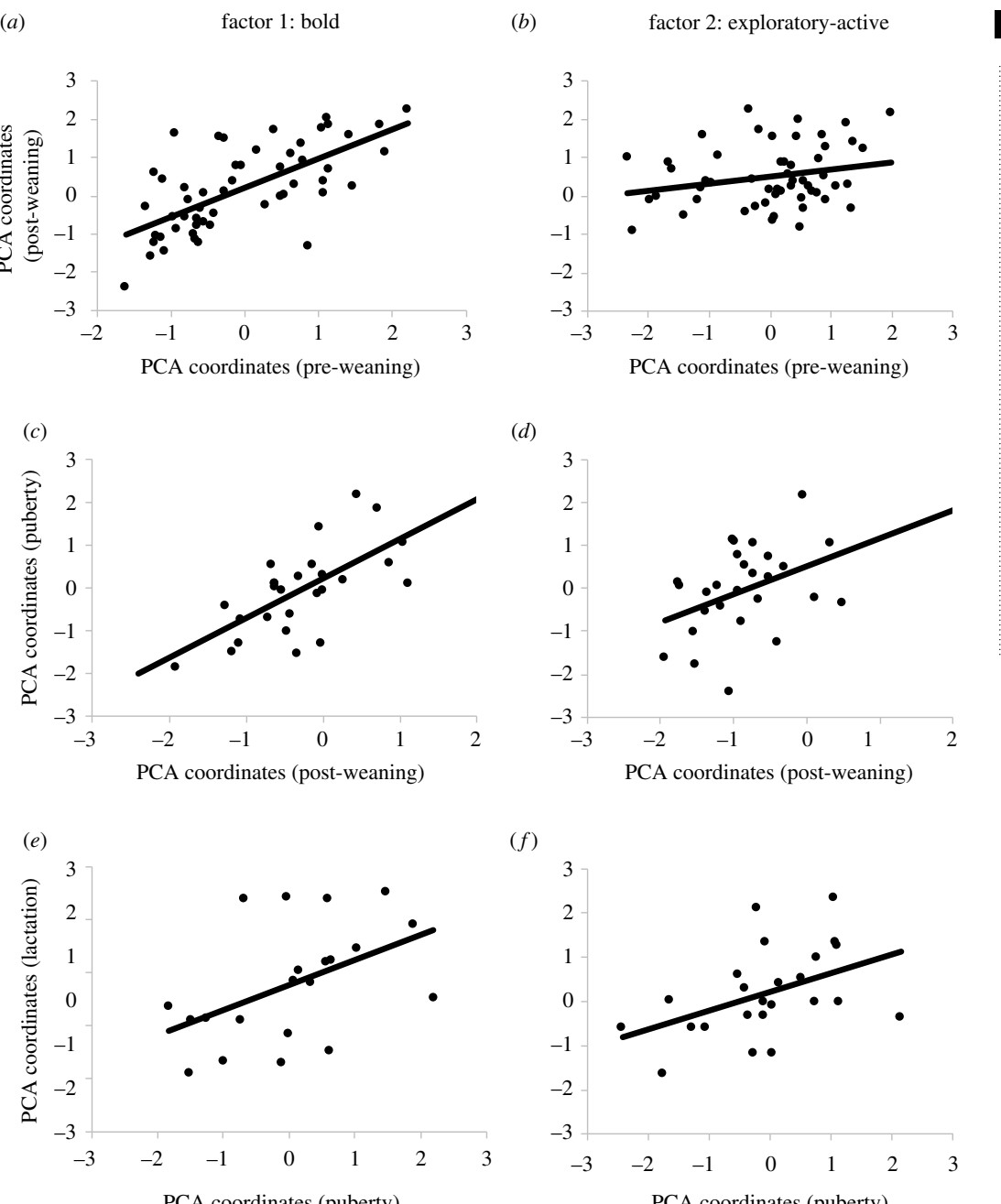

**Figure 1.** Consistency of personality traits of Holstein dairy cattle over time for factor 1 ('boldness' personality trait) and factor 2 ('exploratory/activity' personality trait) between (a,b) pre-weaning and post-weaning (n = 53), (c,d) post-weaning and puberty (n = 27), and (e,f) puberty and lactation periods (n = 21). Individual coordinates reflect the factor loadings for each animal extracted from the PCA. The inverse of coordinate values for factor 1 and factor 2 for post-weaning and lactation periods were used for ease of visualization and interpretation. Therefore, higher values along the x- or y-axis of factor 1 indicate that the animal scored higher for 'boldness' in that period (i.e. fast to make contact with human or object, spent more time touching and playing with human or object). Higher values along the x- or y-axis of factor 2 indicate the animal scored higher for exploration or activity at that period (i.e. spent more time exploring arena, greater movement around arena).

structural consistency of this personality trait. The 'exploration–activity' trait had more moderate structural consistency over time, because high loadings were typically observed for at least two of the exploratory, locomotion, inactivity or attentive behaviours, but not necessarily the same behaviours across all periods. Foris et al. [21] found moderate stability of 'boldness/shyness' trait, and weak stability of 'exploratory/ activity' trait in dairy cattle over a period of six months during lactation, which may be because the novel

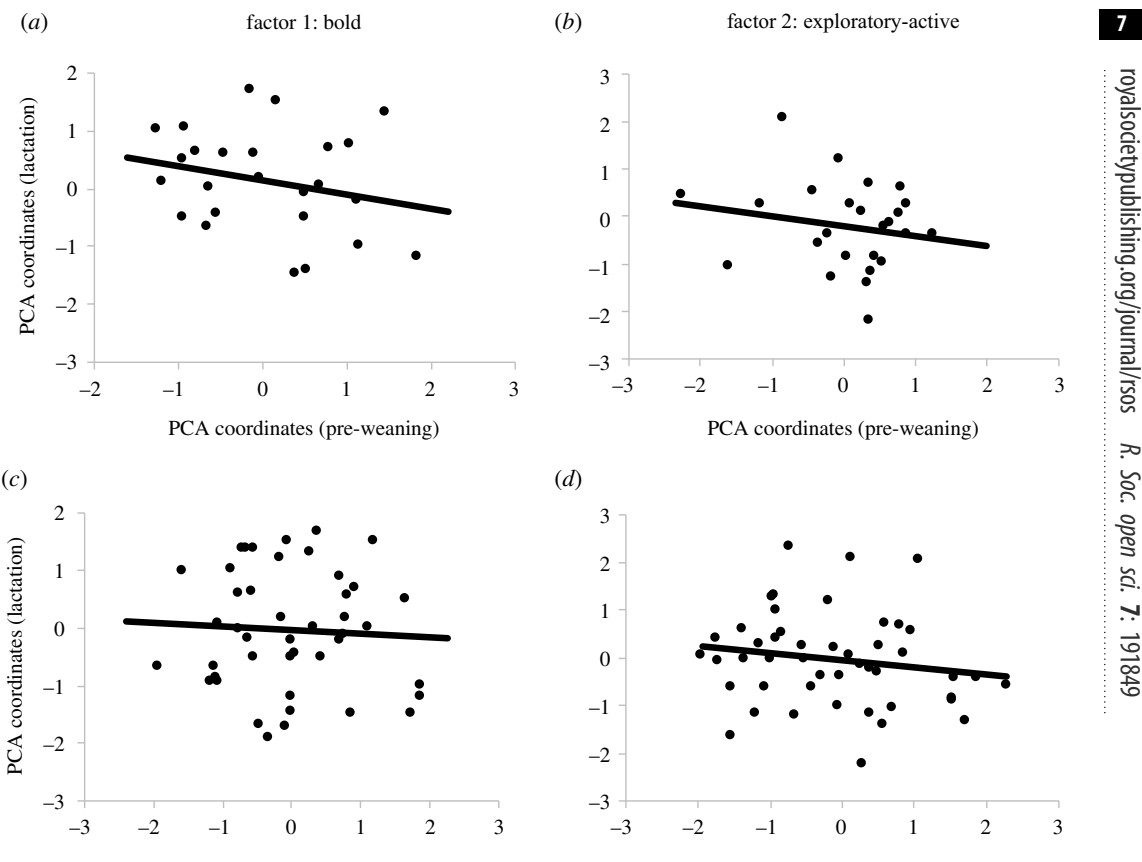

**Figure 2.** Consistency of personality traits of Holstein dairy cattle over time for factor 1 ('boldness' personality trait) and factor 2 ('exploratory/activity' personality trait) between (a,b) pre-weaning and lactation (n = 25), and (d,e) post-weaning and lactation (n = 44). Individual coordinates reflect the factor loadings for each animal extracted from the PCA. The inverse of coordinate values for factor 1 and factor 2 for post-weaning and lactation periods were used for ease of visualization and interpretation. Therefore, higher values along the x- or y-axis of factor 1 indicate that the animal scored higher for 'boldness' in that period (i.e. fast to make contact with human or object, spent more time touching and playing with human or object). Higher values along the x- or y-axis of factor 2 indicate the animal scored higher for exploration or activity at that period (i.e. spent more time exploring arena, greater movement around arena).

environment remained unchanged in their follow-up test. Overall these findings suggest that similar sets of behaviours are correlated when measured multiple times over the first 2 years of life.

There is limited understanding of the development of animal personality traits during ontogenetic change [33], but our findings are in line with several studies in mammals that found instability, especially over major developmental periods such as puberty and sexual maturation (e.g. boldness in marmots [8] and hamsters [34]; exploration in guinea pigs [7] and mice [35]). The mechanisms behind these changes are poorly understood, but changes to the individual's internal state and physical or social environment are likely to also affect how individuals behave, and thus the stability of personality traits [36]. Major physiological changes occur during sexual maturation, which may explain the inconsistency in individual behaviours and personality traits from the juvenile period to the adult period in our study. Steroid hormones around puberty give rise to reproduction-related behaviours typically involving increased risk-taking, exploratory and agonistic behaviours; individual variability in hormone levels has been associated with an exploratory phenotype in great tits [37]. In addition to circulating hormone levels, high metabolic rate, structural size and body mass have been linked with relatively aggressive, bold, exploratory or active personalities, although these intrinsic factors alone did not explain much of the individual variability in behaviour [38]. Growth rate has been associated with differences in personality in female dairy calves [29,39]; growth slows after reaching sexual maturity, which may contribute to greater stability of personality traits at this stage [2]. In combination, this previous work suggests that changes in physiology, metabolism and morphology, which are accelerated during the heifer-rearing period from post-weaning to puberty on

dairy farms, may contribute to fundamental changes in personality structure around puberty. We encourage future work to investigate how these factors impact the development of personality traits.

Two other periods of developmental and physiological change in cattle that we tested were weaning (a period of gastro-intestinal reorganization and rapid development of the rumen) and the onset of lactation (a period of fetal growth, mammary gland development, parturition and milk production). Unlike during puberty and sexual maturation, individual behaviours and personality traits were consistent across these transitions (from pre- to post-weaning, and from puberty to first lactation); we speculate this may be related to the consistency in physical and social environments of our study, especially during the early rearing period when calves remained in the same pen, barn and social group from pre- to post-weaning. Dairy cattle form stable social networks [40]; the stability of these social networks is disrupted when individuals are moved in or out of the group [41]. Mixing of unfamiliar animals after weaning is common practice and causes acute changes in social behaviour [42], which may explain observed changes in behaviour from post-weaning to puberty, and from post-weaning to lactation. The amount of human–animal contact may also contribute to consistency in behaviours and personality traits, where exposure to humans was more limited between post-weaning and puberty periods in our study. Future work is required to understand the specific internal (such as physiological changes) and external (such as social mixing or human–animal contact) factors contributing to changes in behavioural consistency over time.

We found that sex did not affect the expression of boldness or exploratory-active personality traits in dairy calves, or their consistency over time, during the early rearing period. However, we caution that our study included only a small population of male calves that were monitored from pre- to post-weaning, so it is possible that sex differences in personality traits and their consistency over time may exist beyond this period (e.g. as seen during metamorphosis in crickets [43]).

Our findings have applied relevance to the management of commercial dairy cattle. A body of previous work has shown that cattle personality can in part explain variability in feeding behaviour and performance measures on-farm (see review by Neave *et al.* [16]). For instance, cattle that were especially reactive towards novelty or stressful situations also had impaired feeding behaviour [44,45], feed intake [46] and immune function [47]. Cattle that were more reactive when restrained in a crush also had reduced growth [48,49], carcass quality [50] and reproductive performance [51,52]. In dairy cattle specifically, more reactive individuals produced less milk [53,54] and had greater variability in lying behaviour [55]. Most recently, associations between personality traits and performance have also been found in dairy calves [29,39]. Together this work suggests that personality traits in cattle can be a useful indication of how individuals perform on-farm as a calf and as an adult. Given that personality traits were consistent from pre- to post-weaning, and from puberty to first lactation, there is potential to identify individuals at these ages that are most likely to do well or poorly when faced with stressful farm management practices.

# 5. Conclusion

Dairy cattle respond consistently to novelty and have stable personality traits over the early and later periods of development (i.e. from pre- to post-weaning, and from puberty to lactation), but consistency appears to be poor across puberty (i.e. from pre- and post-weaning to lactation). Our findings suggest that personality traits of dairy cattle change over ontogeny, but become more consistent following sexual maturity.

Ethics. The experiment and all procedures were approved by the University of British Columbia's Animal Care Committee (protocol no. A15-0117).

Data accessibility. Raw data and statistical analysis code are included in the electronic supplementary material. Datasets and SAS code can be found in the Mendeley repository at: doi:10.17632/gy7dkst6nm.1.

Authors' contributions. H.W.N., J.H.C.C., D.M.W. and M.A.G.v.K. conceived and designed the study. H.W.N and J.H.C.C. collected and analysed the data. H.W.N. wrote the first draft of the manuscript. J.H.C.C. contributed as secondary author to the manuscript. M.A.G.v.K. and D.M.W. provided supervision, financial support and study materials, interpreted the data, and provided critical feedback on this paper.

Competing interests. The authors declare no competing interests.

Funding. H.W.N. was supported by Canada's Natural Sciences and Engineering Research Council (NSERC) Canada Graduate Scholarship. This work was supported by a Natural Sciences and Engineering Research Council (NSERC) Discovery grant no. NSERC RGPIN-2015-06219 awarded to M.A.G.v.K.

Acknowledgements. We thank all the staff and students of the UBC Dairy Education and Research Centre who helped in this experiment, especially Katrina Rosenberger, Gabriella Marquette, Thomas Ede, Julie Wong, Cheryl Linaksita, Juliana Benetton, Jennifer Van Os, Stephanie Boeve and Benjamin Lecorps for their help with data collection.

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
