## [Reviewer comments · Royal Society Open Science]

Review History

RSOS-191849.R0 (Original submission)

Review form: Reviewer 1 (Vivian Witjes)

Is the manuscript scientifically sound in its present form?

Yes

Are the interpretations and conclusions justified by the results?

Yes

Is the language acceptable?

Yes

Do you have any ethical concerns with this paper?

No

Have you any concerns about statistical analyses in this paper?

Yes

Recommendation?

Accept with minor revision (please list in comments)

Comments to the Author(s)

The paper was highly interesting and I enjoyed learning from it and reviewing it. In the review report you will find detailed suggestions for improvement. For any questions, please do not hesitate to contact me. To summarize (see Appendix A):

- More emphasis on applied relevance may be beneficial;
- Possibly mention that current study is exploring whether cattle personalities are consistent across different life stages, whereas the causes for possible inconsistencies (sexual maturation and disturbances in social stability) may be suggested for future studies;
- Describe most important differences in management for calves between cohort 1 & 2;
- Explain how behaviors scored by the ethogram are transformed to variables for PCAs and how this has led to the final factors: 'exploration/activity' and 'boldness/shyness';
- Explain and provide reference for used >1 eigenvalue threshold;
- Change Pearson correlations in supplementary materials to Spearman Rank correlations;
- Provide statistical procedure for intra-observer reliability analysis;
- Possibly adapt discussion structure;
- Possibly add discussion point of influence of sex differences on personality consistencies;
- Possibly adjust conclusion to summarize only the current results obtained;
- Possibly use consistent terminology to improve clarity.

I recommend that the paper can be accepted as soon as:

- A reference for the >1 eigenvalue threshold for the PCAs is provided;
- Pearson correlations in supplementary materials is changed to Spearman Rank correlations;
- The intra-observer reliability analysis is provided;
- The main differences between management of calves in cohort 1 and 2 are described;
- It is explained how measures from the ethogram are subsequently analysed in the PCAs yielding final 'exploration/activity' and 'boldness/shyness' PCs.

Review form: Reviewer 2

Is the manuscript scientifically sound in its present form?

Yes

Are the interpretations and conclusions justified by the results?

No

Is the language acceptable?

Yes

Do you have any ethical concerns with this paper?

No

Have you any concerns about statistical analyses in this paper?

No

Recommendation?

Accept with minor revision (please list in comments)

Comments to the Author(s)

In this paper the authors describe the consistency of personality traits across ontogeny in dairy cattle from neonate to first lactation as an adult.

In my opinion this is a very interesting job, the method used is correct, as well as the statistical analysis. It is also written very clearly, but nevertheless there are some points that are not understandable to me, as I indicated below.

In my opinion this paper is suitable for publication, after having clarified some points.

ABSTRACT

- Line 19-21: Authors say that: “personality traits may become unstable during stages of ontogeny from infancy to adult, especially during major periods of development such as sexual maturation”. Why did you study bull calves only in pre-weaning and post-weaning? It seems to me that this does not add any useful data to understand the changes caused by sexual maturation.
- Line 26-27: Authors say that they want describe “the consistency of personality traits across ontogeny in dairy cattle from neonate to first lactation as an adult”. So I don’t understand why you include in your study also 24 bull calves, that will never reach first lactation.

INTRODUCTION

- Line 42: Ref. 16, 17: replace with 2, 3.
- Line 51-53: Authors aim to describe development... from calf to adulthood. But bull calves have not been studied until the adult stage, so I don't understand why to include them in the study.

MATERIALS AND METHODS

- Line 80-81: “n = 24 bull calves were studied longitudinally from pre-weaning to first lactation”. 24 bull calves? Were they studied until first lactation?
- Line 92-96: Why Cohort 1 (n = 33 heifer calves) was not studied during pre-weaning? The age of C1 and of C2 were different during post-weaning period of about one month. Why weren't they studied at the same age? At such a young age, even a month's difference can be significant. After puberty the authors have studied only Cohort 2. Why not Cohort 1? The authors say that “personality traits may become unstable during stages of ontogeny from infancy to adult, especially during major periods of development such as sexual maturation”, then why not study the puberty period in Cohort 1? Furthermore, I still do not understand why bull calves have been included in a study that aims to investigate personality changes before weaning to lactation.

RESULTS

- Ok

DISCUSSION

- Line 180-183: the authors say that “consistency of behaviours and personality traits were generally poor across the major developmental period of puberty (from post-weaning to puberty...). This is at odds with what is written in the results, where the authors say that: animals had consistent loadings on the first factor (‘bold’ or ‘shy’)... between puberty and lactation periods, indicating that individual positioning along the boldness-shyness axis was stable during these developmental periods.
- Line 191-192: “inconsistency in individual behaviours and personality traits from the juvenile period to the adult period”. But in results the authors write that Tucker’s congruence coefficient indicated equal similarity for the first factor (‘bold’ or ‘shy’) between pre-weaning and post-weaning (0.98) and between post-weaning and puberty (0.94), and high similarity between puberty and lactation periods (0.90).

Decision letter (RSOS-191849.R0)

05-Dec-2019

Dear Dr Neave,

On behalf of the Editors, I am pleased to inform you that your Manuscript RSOS-191849 entitled "Long-term consistency of personality traits of cattle" has been accepted for publication in Royal

Society Open Science subject to minor revision in accordance with the referee suggestions. Please find the referees' comments at the end of this email.

The reviewers and handling editors have recommended publication, but also suggest some minor revisions to your manuscript. Therefore, I invite you to respond to the comments and revise your manuscript.

- Ethics statement

- Data accessibility

If you wish to submit your supporting data or code to Dryad (<http://datadryad.org/>), or modify your current submission to dryad, please use the following link:
<http://datadryad.org/submit?journalID=RSOS&manu=RSOS-191849>

- Competing interests

- Authors' contributions

- Acknowledgements

- Funding statement

Because the schedule for publication is very tight, it is a condition of publication that you submit the revised version of your manuscript before 14-Dec-2019. Please note that the revision deadline will expire at 00.00am on this date. If you do not think you will be able to meet this date please let me know immediately.

Please note that Royal Society Open Science charge article processing charges for all new submissions that are accepted for publication. Charges will also apply to papers transferred to

Royal Society Open Science from other Royal Society Publishing journals, as well as papers submitted as part of our collaboration with the Royal Society of Chemistry (<https://royalsocietypublishing.org/rsos/chemistry>).

If your manuscript is newly submitted and subsequently accepted for publication, you will be asked to pay the article processing charge, unless you request a waiver and this is approved by Royal Society Publishing. You can find out more about the charges at <https://royalsocietypublishing.org/rsos/charges>. Should you have any queries, please contact openscience@royalsociety.org.

on behalf of Professor Marcelo Sanchez (Associate Editor) and Kevin Padian (Subject Editor)
openscience@royalsociety.org

Editor comments to Author:

I just love the title. Thanks for submitting.

Reviewer comments to Author:

Reviewer: 1
Comments to the Author(s)

The paper was highly interesting and I enjoyed learning from it and reviewing it. In the review report you will find detailed suggestions for improvement. For any questions, please do not hesitate to contact me. To summarize:

- More emphasis on applied relevance may be beneficial;
- Possibly mention that current study is exploring whether cattle personalities are consistent across different life stages, whereas the causes for possible inconsistencies (sexual maturation and disturbances in social stability) may be suggested for future studies;
- Describe most important differences in management for calves between cohort 1 & 2;
- Explain how behaviors scored by the ethogram are transformed to variables for PCAs and how this has led to the final factors: 'exploration/activity' and 'boldness/shyness';
- Explain and provide reference for used >1 eigenvalue threshold;
- Change Pearson correlations in supplementary materials to Spearman Rank correlations;
- Provide statistical procedure for intra-observer reliability analysis;
- Possibly adapt discussion structure;
- Possibly add discussion point of influence of sex differences on personality consistencies;
- Possibly adjust conclusion to summarize only the current results obtained;
- Possibly use consistent terminology to improve clarity.

I recommend that the paper can be accepted as soon as:

- A reference for the >1 eigenvalue threshold for the PCAs is provided;

- Pearson correlations in supplementary materials is changed to Spearman Rank correlations;
- The intra-observer reliability analysis is provided;
- The main differences between management of calves in cohort 1 and 2 are described;
- It is explained how measures from the ethogram are subsequently analysed in the PCAs yielding final 'exploration/activity' and 'boldness/shyness' PCs.

Reviewer: 2

Comments to the Author(s)

In this paper the authors describe the consistency of personality traits across ontogeny in dairy cattle from neonate to first lactation as an adult.

In my opinion this is a very interesting job, the method used is correct, as well as the statistical analysis. It is also written very clearly, but nevertheless there are some points that are not understandable to me, as I indicated below.

In my opinion this paper is suitable for publication, after having clarified some points.

ABSTRACT

- Line 19-21: Authors say that: “personality traits may become unstable during stages of ontogeny from infancy to adult, especially during major periods of development such as sexual maturation”. Why did you study bull calves only in pre-weaning and post-weaning? It seems to me that this does not add any useful data to understand the changes caused by sexual maturation.
- Line 26-27: Authors say that they want describe “the consistency of personality traits across ontogeny in dairy cattle from neonate to first lactation as an adult”. So I don’t understand why you include in your study also 24 bull calves, that will never reach first lactation.

INTRODUCTION

- Line 42: Ref. 16, 17: replace with 2, 3.
- Line 51-53: Authors aim to describe development... from calf to adulthood. But bull calves have not been studied until the adult stage, so I don't understand why to include them in the study.

MATERIALS AND METHODS

- Line 80-81: “n = 24 bull calves were studied longitudinally from pre-weaning to first lactation”. 24 bull calves? Were they studied until first lactation?
- Line 92-96: Why Cohort 1 (n = 33 heifer calves) was not studied during pre-weaning? The age of C1 and of C2 were different during post-weaning period of about one month. Why weren't they studied at the same age? At such a young age, even a month's difference can be significant. After puberty the authors have studied only Cohort 2. Why not Cohort 1? The authors say that “personality traits may become unstable during stages of ontogeny from infancy to adult, especially during major periods of development such as sexual maturation”, then why not study the puberty period in Cohort 1? Furthermore, I still do not understand why bull calves have been included in a study that aims to investigate personality changes before weaning to lactation.

RESULTS

- Ok

DISCUSSION

- Line 180-183: the authors say that “consistency of behaviours and personality traits were generally poor across the major developmental period of puberty (from post-weaning to puberty...). This is at odds with what is written in the results, where the authors say that: animals

had consistent loadings on the first factor ('bold' or 'shy')... between puberty and lactation periods, indicating that individual positioning along the boldness-shyness axis was stable during these developmental periods.

- Line 191-192: "inconsistency in individual behaviours and personality traits from the juvenile period to the adult period". But in results the authors write that Tucker's congruence coefficient indicated equal similarity for the first factor ('bold' or 'shy') between pre-weaning and post-weaning (0.98) and between post-weaning and puberty (0.94), and high similarity between puberty and lactation periods (0.90).

Author's Response to Decision Letter for (RSOS-191849.R0)

See Appendix B.

Decision letter (RSOS-191849.R1)

15-Jan-2020

Dear Dr Neave,

It is a pleasure to accept your manuscript entitled "Long-term consistency of personality traits of cattle" in its current form for publication in Royal Society Open Science. The comments of the reviewer(s) who reviewed your manuscript are included at the foot of this letter.

Per your request to update the corresponding author to Dr Marina von Keyserlingk, I have made this change: Marina will be the primary contact during the proofing process.

Additionally, please can you ensure that your Mendeley dataset (10.17632/gy7dkst6nm.1) is active before publication of the paper? We will ensure this link is included in the proofed paper.

on behalf of Professor Marcelo Sanchez (Associate Editor) and Kevin Padian (Subject Editor)
openscience@royalsociety.org

Follow Royal Society Publishing on Twitter: [@RSocPublishing](https://twitter.com/RSocPublishing)
Follow Royal Society Publishing on Facebook:
<https://www.facebook.com/RoyalSocietyPublishing.FanPage/>
Read Royal Society Publishing's blog: <https://blogs.royalsociety.org/publishing/>

A

Peer review report RSOS-191849 LONG-TERM CONSISTENCY OF PERSONALITY TRAITS OF CATTLE

Authors: Heather W. Neave, Joao H. C. Costa, Daniel M. Weary, and Marina A. G. von Keyserlingk
Referee: Vivian L. Witjes (vivian.witjes@vetsuisse.unibe.ch)

Article summary

Recent studies have shown that personality traits in farmed ruminants may be correlated to individual health, productivity and welfare. Additionally, in several species from various taxa, personality changes and inconsistency of certain traits across the life-span have been observed. However, in domesticated dairy cattle personality instability has not yet been investigated, even though this might affect their performance and quality of life. Therefore, the aim of this study was to assess the personality consistency of dairy cattle by testing animals at four different life stages, namely during pre-weaning, post-weaning, after puberty and during adult life. Behavioural responses to a novel environment-, novel human-, and novel object task were quantified at each life stage and the consistency of behavioural traits across three years were analysed. It was found that behavioural signs of "boldness/shyness" and "exploration/activity" were mostly consistent from pre- to post-weaning and from puberty to lactation. In contrast, these traits were more inconsistent from pre- and post-weaning to lactation, suggesting dairy cows undergo changes in personality due to physiological (mainly hormonal) changes during sexual maturation.

First impressions

Only very recently have studies focused on animal personality consistency emerged. Since no research on personality stability across the different life phases of dairy cattle have been conducted, this is the first study to offer insight into critical periods for when changes in personality might occur. Hence, this study increases our fundamental knowledge on animal (specifically cattle) personality and may additionally be valuable to the applied field of cattle farming, since management practices may need to be adapted during these critical periods to promote productivity and welfare. However, the paper could benefit from an increased emphasis on the applied relevance and including interpretations for what the obtained results mean for cattle farming. The article design and language are sufficient and the general directive of the study is clear.

Title and abstract

The title could benefit from more detail of the current study and obtained results, since the title could now be interpreted as a review article (e.g. 'Assessing long-term consistency of personality traits of dairy cattle' or 'Personality inconsistencies during sexual maturation of dairy cattle'). The abstract is concise and informative of the background, although a description of the used methodology and statistical results are lacking.

Introduction

The introduction is well-written and clear, clarifying the background of the study as well as its aim. No specific hypotheses are stated, but the research question is clear. As previously mentioned, the applied relevance of the study may possibly be emphasized more. In addition, the last sentence of the introduction states that the current study 'may offer insight into the factors affecting stability of personality across ontogeny', while the nature of the study is rather preliminary/exploratory than specifically assessing factors causing personality changes. Due to the experimental set-up (e.g. no physiological measures or social behaviours are included as parameters), effects of physiological changes (e.g. sexual maturation) and management practices (e.g. social regrouping) on personality consistency cannot be disentangled and the results show during what life phases instability of personality traits is likely to occur. Therefore, a more cautious rephrasing could be considered.

Methodology

Most of the used methods are well described, although some information is lacking. No detailed description of the differences in management during early life of cohort 1 and 2 is given. Therefore, it remains unclear whether this may have affected the development and stability of personality traits in calves. In addition, the parameters measured by behavioural observations using the ethogram (table S1) during all tests (i.e. 'locomotor play', 'bucking', 'resting' and 'withdrawal') seem to be excluded from all analyses, while no reasons are stated for why these behaviours were measured and subsequently excluded. Alternatively, these behaviours were combined with the test-specific behavioural measures (e.g. 'locomotor play' combined with 'activity' and 'resting' with 'inactivity' during novel environment test), though this is not mentioned.

Furthermore, the reason why not all variables are included in the principle component analyses (PCAs) is a little unclear. My interpretation is that, the variables chosen as input parameters for the PCAs are based on the Spearman rank correlations, which should be clarified by specifically stating this, yielding the final 'boldness/shyness' and 'exploration/activity' factors. The overall methodology is referenced to relevant articles, although a reference for the >1 eigenvalue threshold for including principal components is missing. Since the variation explained by the two first PCs ranges from 55.4% to 60.9% across all data for different life stages, authors should consider including more principal components for further analysis, because they could contribute substantially to the variation observed (e.g. for the data gathered during the lactation period, a third PC with an eigenvalue of 0.8 should in this case still be considered, since the first two factors only explain approximately half of the variation). If the authors decide not to include additional PCs, a reference for the eigenvalue threshold is needed to clarify why the current methodology was chosen, which can aid researchers inexperienced in PCA analysis such as myself in understanding the statistical methods.

Finally, the statistical procedure for how inter-observer reliability was obtained is lacking and it is unclear whether this was measured just once or multiple times during the study, to ensure that behaviours were reliably scored throughout all life stages of the animals. The supplementary materials are informative and clear, besides some previously mentioned shortcomings. Moreover, in the supplementary methods, it is mentioned that Pearson correlations were performed, whereas in the text and table 1, it is stated that Spearman rank correlations were conducted, thus specification of which test was used is needed. In the statistical code, it is stated that personality consistency between pre-weaning and puberty is not analysed, because none of the animals tested during pre-weaning were additionally tested during puberty (although all animals tested during both periods originate from cohort 2). This should be included in the supplementary materials as clarification for why this comparison is not reported.

Results and discussion

The results are clear and well-explained and no results are missing. However, changing the term for the second PC to 'exploration/inactivity' may be more consistent with the first PC, since the first PC term ('boldness/shyness') indicates the extremes of the component's axis. Additionally, the discussion provides an in-depth interpretation of the results in relation to the existing literature. Nevertheless, the article may benefit from altering the structure of the discussion, so that the specific interpretation of the current results are discussed first, before considering a more general discussion (i.e. move lines 222 to 237 before line 184). This way, the paper will follow an hourglass-shaped structure from introduction to conclusion, guiding readers from the more general line of thought to specific hypothesis tested, on to the specific obtained results, ending with a general interpretation of these results.

A possible additional discussion point is that both sexes are included in one of the cohorts and participate in tests during the pre- and post-weaning stages of the study, after which all males are excluded. Since the study is focused on measuring personality traits and consistency during different life phases, sex may be an important confounding factor, since males and females may differ in their personality traits and consistency across development due to different hormonal/physiological changes. Although males are only included in the early life stages and sex may then not have a substantial effect, this issue could still be considered.

Furthermore, it is mentioned that, especially after weaning, group mixing can occur following standard management practices. Thus, since the results indicate that personality inconsistencies in female dairy cattle may arise during puberty, the authors could suggest a focus of future research on assessing whether personality changes are caused by physiological changes during sexual maturation or social group mixing (or other management-related environmental changes) or a combination of both.

Conclusion

The final conclusions are concise and are valid judging from the results obtained. However, since the current study has not investigated the causes of personality inconsistency during puberty, the comment that the results are likely due to physiological changes is a little preliminary, since they could alternatively be caused by environmental factors (e.g. social regrouping). Nevertheless, the overall results allow for an increased knowledge about dairy cattle personality, which is summarised by the conclusions.

References, tables and figures

To my knowledge, the references are relevant and referenced correctly. A possible additional reference contributing to how behavioural tests can be used as a proxy for animal personality is:

"Koski S.E. (2011) How to Measure Animal Personality and Why Does It Matter? Integrating the Psychological and Biological Approaches to Animal Personality. In: Inoue-Murayama M., Kawamura S., Weiss A. (eds) From Genes to Animal Behavior. Primatology Monographs. p115-136, Springer, Tokyo."

The tables, figures and associated descriptions provided are informative and aid in understanding and interpreting the methods and results. A suggestion for improvement is consistency of terminology. Table 1 displays correlations of individual behavioural measures in each of the tests across periods and table S1 presents the ethogram used for behavioural observations. Clarifying that, for example, 'licking/sniffing wall or floor' (table S1) is a measure of 'exploration' (table 1) by using the same terms for the same measurement can prevent confusion. Similarly, the terms 'puberty' and 'lactation' are used in the text to refer to two of the periods during which tests were performed, whereas in the figures (figure 1 and 2) these periods are indicated by the terms 'heifer' and 'cow', respectively.

Recommendation

The article gives me no reason to suspect plagiarism, fraud or other ethical issues. I recommend that the paper is accepted, however minor revision is needed, as commented above. If necessary, I am willing to go over any adjustments made by the authors.

B

Response to Reviewers (Revision 1): Long-term consistency of personality traits of cattle

Editor comments to Author:

I just love the title. Thanks for submitting.

AU: Thank you!

Reviewer comments to Author:

Reviewer: 1

Comments to the Author(s)

The paper was highly interesting and I enjoyed learning from it and reviewing it. In the review report you will find detailed suggestions for improvement. For any questions, please do not hesitate to contact me.

AU: Thank you very much, we are happy to hear the reviewer found our paper interesting to read.

To summarize:

- More emphasis on applied relevance may be beneficial;

AU: we have added a paragraph at the end of the discussion to address application of our findings on L266-279: "Our findings have applied relevance to the management of commercial dairy cattle. A body of previous work has shown that cattle personality can account for variability in feeding behaviour and performance measures on-farm (see review by Neave et al. [16]). For instance, cattle that were especially reactive toward novelty or stressful situations also had impaired feeding behaviour [44,45], feed intake [46], and immune function [47]. Cattle that were more reactive when restrained in a crush also had reduced growth rates [48,49], carcass quality [50] and reproductive performance [51,52]. In dairy cattle specifically, more reactive individuals produced less milk [53,54] and had greater variability in lying behaviour [55]. Most recently, associations between personality traits and performance have also been found in dairy calves [29,37]. Together this work suggests that the measurement of personality traits in cattle can be a useful indication of how individuals perform on-farm as a calf or as an adult. Given that personality traits were consistent from pre- to post-weaning, and from puberty to first lactation, there is potential to identify individuals at these ages that are most likely to do well or poorly when faced with stressful farm management practices."

- Possibly mention that current study is exploring whether cattle personalities are consistent across different life stages, whereas the causes for possible inconsistencies

(sexual maturation and disturbances in social stability) may be suggested for future studies;

AU: We have clarified this line of future work in the introduction L68: “The study of behavioural patterns and personality traits across these transitions may offer insight into the factors affecting stability of personality across ontogeny.”
And in the discussion L240: “In combination, this previous work suggests that changes in physiology, metabolism and morphology, which are accelerated during the heifer-rearing period from post-weaning to puberty on dairy farms, may contribute to fundamental changes in personality structure around puberty. We encourage future work to investigate how these factors may impact development of personality traits.”

- Describe most important differences in management for calves between cohort 1 & 2;

AU: We have described these differences in more detail in both the manuscript (L80) and supplementary material: “All animals were reared on the same farm and management of animals was the same for all periods except for the pre-weaning period. Calves in Cohort 1 (born November 2014 to Feb 2015) received 12 L/d of milk for 6 wk then weaned over 2 wk to complete weaning at 8 wk, while calves in Cohort 2 received different milk allowances (6, 8, 10 or 12 L/d of milk) and weaned on a step-down weaning program reported in Neave et al. [1] (milk allowance did not affect measures of personality, see Neave et al. [1]).”

- Explain how behaviors scored by the ethogram are transformed to variables for PCAs and how this has led to the final factors: 'exploration/activity' and 'boldness/shyness';

AU: This information was provided in the supplementary material, but we have provided more details regarding the PCA analysis and criteria. “PCA analysis criteria and calculations followed the recommendations outlined by Comrey and Lee [3] and Budaev [4]. For each PCA, the correlation matrix was computed (see Supplementary Tables S4-7) and the first two factors were retained (eigenvalues > 1 and following scree plot examination) and subjected to orthogonal (varimax) rotation. The variables used for analysis met the criteria of Kaiser–Meyer–Olkin measure of sampling adequacy (Pre-weaning = 0.72, Post-weaning = 0.67, Puberty = 0.68, Lactation = 0.47) required for conducting PCA. Community estimates were adequate for most variables; a variable that was $h^2 < 0.40$ in one of the periods was $h^2 > 0.40$ in another period and thus this variable was retained in the PCA for all periods (Pre-weaning > 0.41, Post-weaning > 0.28, Puberty > 0.01, Lactation > 0.34). Individual scores on each of the PCA factors were extracted using the regression method. A variable was considered to have a high loading on a factor if $> \pm 0.62$.”

- Explain and provide reference for used >1 eigenvalue threshold;

AU: This is now reported in the supplementary material: “PCA analysis criteria and calculations followed the recommendations outlined by Comrey and Lee [3] and Budaev [4].”

- Change Pearson correlations in supplementary materials to Spearman Rank correlations;

AU: If the reviewer is referring to the Pearson correlations performed between the factor scores of the PCAs, then we chose this correlation to be Pearson since the factor loadings extracted from the PCA are normally distributed.

- Provide statistical procedure for intra-observer reliability analysis;

AU: This procedure was Cohen's kappa and is now reported in the supplementary material: "A single observer scored all behaviors in all tests following Neave et al. [1] (see Supplementary Table S1) after achieving sufficient inter-observer reliability (Cohen's kappa $\kappa > 0.80$)."

- Possibly adapt discussion structure;

AU: We have adjusted the discussion structure as outlined in the additional comments provided via PDF, which included moving a paragraph closer to the beginning of the discussion (now L208-223). We have also added two new discussion points at the end of the discussion, following suggestions from both reviewers (L262-281).

- Possibly add discussion point of influence of sex differences on personality consistencies;

AU: We have now investigated the effect of sex on expression of personality traits, and their consistency over time. We found no sex differences during the pre- and post-weaning period. We have included these analyses in the statistics section (L125-132), reported in a separate section of the results (L190-195), and a short paragraph in the discussion about the findings (L262-267).

- Possibly adjust conclusion to summarize only the current results obtained;

AU: We removed the speculation regarding why personality may change over puberty. The conclusion now strictly focuses on the findings.

- Possibly use consistent terminology to improve clarity.

AU: We have adjusted our terminology to refer to only boldness and exploratory traits (rather than boldness-shyness, and exploratory-activity). We have therefore reversed the factor loadings on some of the Factors for some of periods to align with this terminology use, which we hope improves clarity: L158-160 "For ease of interpretation and to be consistent with literature terminology, the loadings on some factors were reversed (Factor 1 at post-weaning and puberty periods; Factor 2 at puberty and lactation periods)." We also provided definitions for personality traits in the introduction on L50-55: "The most common personality traits investigated in farm animals include 'boldness' (i.e. the propensity to

take risks when exposed to novelty [17]), 'exploration' (i.e. behavior that gains information about the environment [18]) and 'sociability' (i.e. propensity to seek contact with or remain close to conspecifics [1]); these traits are typically inferred by observing behavioural responses toward novelty or isolation in standardized tests [19]."

I recommend that the paper can be accepted as soon as:

- A reference for the >1 eigenvalue threshold for the PCAs is provided;
- Pearson correlations in supplementary materials is changed to Spearman Rank correlations;
- The intra-observer reliability analysis is provided;
- The main differences between management of calves in cohort 1 and 2 are described;
- It is explained how measures from the ethogram are subsequently analysed in the PCAs yielding final 'exploration/activity' and 'boldness/shyness' PCs.

AU: We have addressed each of these points (see responses to each above).

Reviewer: 2

Comments to the Author(s)

In this paper the authors describe the consistency of personality traits across ontogeny in dairy cattle from neonate to first lactation as an adult.

In my opinion this is a very interesting job, the method used is correct, as well as the statistical analysis. It is also written very clearly, but nevertheless there are some points that are not understandable to me, as I indicated below.

In my opinion this paper is suitable for publication, after having clarified some points.

AU: Thank you!

ABSTRACT

- Line 19-21: Authors say that: "personality traits may become unstable during stages of ontogeny from infancy to adult, especially during major periods of development such as sexual maturation". Why did you study bull calves only in pre-weaning and post-weaning? It seems to me that this does not add any useful data to understand the changes caused by sexual maturation.

AU: The specific aim of the study was to investigate consistency of personality traits of dairy cattle over time. We had the opportunity to include bull calves for one of the cohorts, but due to farm management we were unable to keep these calves beyond the post-weaning period. Bull calves may remain on dairy farms for the first stage of rearing – therefore, an understanding of bull calf personalities and their consistency during this early rearing period is also relevant. We have now investigated the effect of sex on expression of personalities (i.e. factor loadings on Factor 1 and Factor 2) and consistency over the pre-weaning to post-weaning period. We have included a line at

the end of the introduction (L70) to more clearly describe the inclusion of the bull calves for the early rearing period: “Our study also included a sub-sample of male calves that were only studied during the early rearing period; this provided an opportunity to investigate sex differences in the expression of personality traits in dairy calves. “

- Line 26-27: Authors say that they want describe “the consistency of personality traits across ontogeny in dairy cattle from neonate to first lactation as an adult”. So I don’t understand why you include in your study also 24 bull calves, that will never reach first lactation.

AU: See above – we have clarified the use of the bull calves at end of the introduction and in the methods section.

INTRODUCTION

- Line 42: Ref. 16, 17: replace with 2, 3.

AU: Corrected

- Line 51-53: Authors aim to describe development.... from calf to adulthood. But bull calves have not been studied until the adult stage, so I don’t understand why to include them in the study.

AU: See response above – clarification for bull calf use is provided in introduction on L70: “Our study also included a sub-sample of male calves that were only studied during the early rearing period; this provided an opportunity to investigate sex differences in the expression of personality traits in dairy calves.”

MATERIALS AND METHODS

- Line 80-81: “n = 24 bull calves were studied longitudinally from pre-weaning to first lactation”. 24 bull calves? Were they studied until first lactation?

AU: No the bull calves were sold after post-weaning and so were unable to be tested for later developmental periods. This has been outlined in the text (L87): “Some female calves and all male calves were sold following farm management procedures after the post-weaning period and thus were unavailable for final testing...”

- Line 92-96: Why Cohort 1 (n = 33 heifer calves) was not studied during pre-weaning? The age of C1 and of C2 were different during post-weaning period of about one month. Why weren't they studied at the same age? At such a young age, even a month's difference can be significant. After puberty the authors have studied only Cohort 2. Why not Cohort 1? The authors say that “personality traits may become unstable during stages of ontogeny from infancy to adult, especially during major periods of development such as sexual maturation”, then why not study the puberty period in Cohort 1? Furthermore, I still do not understand why bull calves have been

included in a study that aims to investigate personality changes before weaning to lactation.

AU: Cohort 1 was a convenience sample that was only enrolled in the post-weaning period when researchers arrived at the experimental farm. Cohort 2 was not tested in the puberty period due to time constraints and the lack of available space in the barn for testing. We have clarified this in the text (L97-99) “Cohort 1 was not tested in the pre-weaning period (due to enrollment in the study at post-weaning only), and Cohort 2 was not tested in the puberty period (due to time and barn space constraints).”

The age difference between Cohort 1 and 2 for post-weaning period was due to time of enrollment for Cohort 1 and timing of personality testing for another experiment for Cohort 2 (reported in Neave et al., 2018. J. Dairy Sci.). Unfortunately we do not have a large range of ages to be able to properly test for an effect of age at testing in the post-weaning period, but we would not expect this to impact expression of personality traits (especially given that scores were consistent from pre-weaning to post-weaning period).

RESULTS

- Ok

DISCUSSION

- Line 180-183: the authors say that “consistency of behaviours and personality traits were generally poor across the major developmental period of puberty (from post-weaning to puberty...). This is at odds with what is written in the results, where the authors say that: animals had consistent loadings on the first factor (‘bold’ or ‘shy’)... between puberty and lactation periods, indicating that individual positioning along the boldness-shyness axis was stable during these developmental periods.

AU: Thank you for pointing out this error. We have removed reference to the post-weaning to lactation period and now state (L204-206) “However, individual consistency of behaviours and personality traits were generally poor across the major developmental period of puberty (from pre-weaning to lactation, and from post-weaning to lactation).”

- Line 191-192: “inconsistency in individual behaviours and personality traits from the juvenile period to the adult period”. But in results the authors write that Tucker’s congruence coefficient indicated equal similarity for the first factor (‘bold’ or ‘shy’) between pre-weaning and post-weaning (0.98) and between post-weaning and puberty (0.94), and high similarity between puberty and lactation periods (0.90).

AU: The Tucker’s congruence is identifying similarity in overall dimensions at each period, not how each individual scores on these Factors; we have clarified this in the statistics section (L122): “Tucker’s coefficient of congruence was also performed as a measure of factor similarity (i.e. similar correlated sets of behaviours across periods, and thus structural consistency) across periods (following [30]).”

And results section (L182): “Tucker’s congruence coefficient indicated equal similarity of overall loadings (i.e. correlated sets of behaviours) on the first factor (‘bold’) between pre-weaning and post-weaning (0.98)”

Additional comments from Peer Review Report from Vivian L. Witjes
(vivian.witjes@vetsuisse.unibe.ch)

Title and abstract:

The title could benefit from more detail of the current study and obtained results, since the title could now be interpreted as a review article (e.g. 'Assessing long-term consistency of personality traits of dairy cattle' or 'Personality inconsistencies during sexual maturation of dairy cattle'). The abstract is concise and informative of the background, although a description of the used methodology and statistical results are lacking.

AU: Thank you for the suggestion. We prefer to keep a concise title that purposely refrains from stating consistency or inconsistency given this refers to specific time points.

Introduction

Due to the experimental set-up (e.g. no physiological measures or social behaviours are included as parameters), effects of physiological changes (e.g. sexual maturation) and management practices (e.g. social regrouping) on personality consistency cannot be disentangled and the results show during what life phases instability of personality traits is likely to occur. Therefore, a more cautious rephrasing could be considered.

AU: We have rephrased to be clear we are not testing the influence of these factors on L68: “The study of behavioural patterns and personality traits across these transitions may offer insight into the factors affecting stability of personality across ontogeny, although we do not aim to specifically test the influence of each of these factors.”

Methods:

In addition, the parameters measured by behavioural observations using the ethogram (table S1) during all tests (i.e. 'locomotor play', 'bucking', 'resting' and 'withdrawal') seem to be excluded from all analyses, while no reasons are stated for why these behaviours were measured and subsequently excluded. Alternatively, these behaviours were combined with the test-specific behavioural measures (e.g. 'locomotor play' combined with 'activity' and 'resting' with 'inactivity' during novel environment test), though this is not mentioned.

AU: We have clarified in supplementary material that the variables bucking and withdrawals have been excluded due to few occurrences: “Frequencies of bucking and withdrawals were too few to be meaningfully included in the analysis.”

My interpretation is that, the variables chosen as input parameters for the PCAs are based on the Spearman rank correlations, which should be clarified by specifically stating this, yielding the final 'boldness/shyness' and 'exploration/activity' factors. The overall methodology is referenced

to relevant articles, although a reference for the >1 eigenvalue threshold for including principal components is missing. Since the variation explained by the two first PCs ranges from 55.4% to 60.9% across all data for different life stages, authors should consider including more principal components for further analysis, because they could contribute substantially to the variation observed (e.g. for the data gathered during the lactation period, a third PC with an eigenvalue of 0.8 should in this case still be considered, since the first two factors only explain approximately half of the variation). If the authors decide not to include additional PCs, a reference for the eigenvalue threshold is needed to clarify why the current methodology was chosen, which can aid researchers inexperienced in PCA analysis such as myself in understanding the statistical methods.

AU: We included all variables except the 2 with too few frequencies as reported now. In the supplementary material, we have provided reference to articles for PCA criteria and the criterion for the eigenvalue of > 1 . We also report further metrics to provide assurances that PCA analysis was appropriate for this data set.

Results and Discussion

The results are clear and well-explained and no results are missing. However, changing the term for the second PC to 'exploration/inactivity' may be more consistent with the first PC, since the first PC term ('boldness/shyness') indicates the extremes of the component's axis. Additionally, the discussion provides an in-depth interpretation of the results in relation to the existing literature. Nevertheless, the article may benefit from altering the structure of the discussion, so that the specific interpretation of the current results are discussed first, before considering a more general discussion (i.e. move lines 222 to 237 before line 184). This way, the paper will follow an hourglass-shaped structure from introduction to conclusion, guiding readers from the more general line of thought to specific hypothesis tested, on to the specific obtained results, ending with a general interpretation of these results.

AU: Thank you for this suggestion – we have moved the section up so that this paragraph now falls on L208-223. Therefore as suggested, we discuss the specific results first and then move into a more general discussion.

A possible additional discussion point is that both sexes are included in one of the cohorts and participate in tests during the pre- and post-weaning stages of the study, after which all males are excluded. Since the study is focused on measuring personality traits and consistency during different life phases, sex may be an important confounding factor, since males and females may differ in their personality traits and consistency across development due to different hormonal/physiological changes. Although males are only included in the early life stages and sex may then not have a substantial effect, this issue could still be considered.

AU: Thank you for this valid point. We have included analysis and short discussion of sex in our manuscript now (as described in earlier comments).

Furthermore, it is mentioned that, especially after weaning, group mixing can occur following standard management practices. Thus, since the results indicate that personality inconsistencies

in female dairy cattle may arise during puberty, the authors could suggest a focus of future research on assessing whether personality changes are caused by physiological changes during sexual maturation or social group mixing (or other management-related environmental changes) or a combination of both.

AU: We have suggested some of these specific factors to investigate at the end of our discussion on L259: “Future work is required to understand the specific internal (such as physiological changes) and external (such as social mixing or human-animal contact) factors contributing to changes in behavioural consistency over time.”